# The Upconversion Luminescence of Ca_3_Sc_2_Si_3_O_12_:Yb^3+^,Er^3+^ and Its Application in Thermometry

**DOI:** 10.3390/nano13131910

**Published:** 2023-06-22

**Authors:** Junyu Hong, Feilong Liu, Miroslav D. Dramićanin, Lei Zhou, Mingmei Wu

**Affiliations:** 1School of Chemical Engineering and Technology, Sun Yat-sen University, Zhuhai 519082, China; hongjy23@mail.sysu.edu.cn (J.H.); ceswmm@mail.sysu.edu.cn (M.W.); 2School of Marine Sciences, Sun Yat-sen University, Zhuhai 519082, China; liuflong3@mail2.sysu.edu.cn; 3Center of Excellence for Photoconversion, Vinča Institute of Nuclear Sciences-National Institute of the Republic of Serbia, University of Belgrade, PO Box 522, 11001 Belgrade, Serbia; dramican@vinca.rs

**Keywords:** luminescence thermometry, upconversion luminescence, phosphor, lanthanides, sol–gel synthesis

## Abstract

To develop novel luminescent materials for optical temperature measurement, a series of Yb^3+^- and Er^3+^-doped Ca_3_Sc_2_Si_3_O_12_ (CSS) upconversion (UC) phosphors were synthesized by the sol–gel combustion method. The crystal structure, phase purity, and element distribution of the samples were characterized by powder X-ray diffraction and a transmission electron microscope (TEM). The detailed study of the photoluminescence emission spectra of the samples shows that the addition of Yb^3+^ can greatly enhance the emission of Er^3+^ by effective energy transfer. The prepared Yb^3+^ and Er^3+^ co-doped CSS phosphors exhibit green emission bands near 522 and 555 nm and red emission bands near 658 nm, which correspond to the ^2^H_11/2_→^4^I_15/2_, ^4^S_3/2_→^4^I_15/2_, and ^4^F_9/2_→^4^I_15/2_ transitions of Er^3+^, respectively. The temperature-dependent behavior of the CSS:0.2Yb^3+^,0.02Er^3+^ sample was carefully studied by the fluorescence intensity ratio (FIR) technique. The results indicate the excellent sensitivity of the sample, with a maximum absolute sensitivity of 0.67% K^−1^ at 500 K and a relative sensitivity of 1.34% K^−1^ at 300 K. We demonstrate here that the temperature measurement performance of FIR technology using the CSS:Yb^3+^,Er^3+^ phosphor is not inferior to that of infrared thermal imaging thermometers. Therefore, CSS:Yb^3+^,Er^3+^ phosphors have great potential applications in the field of optical thermometry.

## 1. Introduction

As a promising remote temperature measurement method, fluorescence intensity ratio (FIR) technology has attracted extensive attention because of its wide temperature response range, high sensitivity, fast response, and submicron measurement scale compared with traditional contact temperature measurement technology [1,2,3,4,5,6,7,8]. In addition, it can be used in highly corrosive, high-pressure, and internal biological tissues [9]. For instance, Vetrone et al. [10] devised a novel nanothermometer, capable of accurately determining the temperature of biological systems such as HeLa cancer cells. The nanothermometer is based on the temperature-sensitive fluorescence of NaYF_4_:Er^3+^,Yb^3+^ nanoparticles, where the intensity ratio of the green fluorescence bands of Er^3+^ ions changes with temperature. Following incubation of the nanoparticles with HeLa cervical cancer cells and their subsequent uptake, the fluorescent nanothermometer measured the internal temperature of the living cells from 25 °C to 45 °C under the excitation of a 920 nm laser. Optical thermometry based on FIR technology has been widely studied in rare-earth-doped upconversion (UC) phosphors. Generally, FIR technology is realized through the temperature dependence of thermally coupled energy levels (TCL) such as the ^2^H_11/2_ and ^4^S_3/2_ levels of Er^3+^, the ^3^F_2, 3_ and ^3^H_4_ levels of Tm^3+^, and the ^5^F_4_ and ^5^S_2_ levels of Ho^3+^ [11,12,13]. Among them, Er^3+^ is usually used as a luminescence center to detect temperature due to the typical green UC emission, appropriate TCL energy gap (~800 cm^−1^), negligible electromagnetic interference, and wide dynamic range. However, the small absorption cross-section of Er^3+^ in the near-infrared region results in low UC emission efficiency. Therefore, Yb^3+^, which has a larger absorption cross-section and can effectively transfer energy to Er^3+^, is often used as a sensitizer to improve the UC efficiency of Er^3+^.

It is well known that hosts with low phonon energy can provide high UC emission efficiency by inhibiting multiphoton relaxation. Fluoride compounds with low phonon energy are often used as host materials of UC phosphors [14]. Fan et al. [15] prepared Li^+^ co-doped *β*-NaYF_4_:Yb^3+^,Er^3+^ micro-cylindrical particles with high UC quantum efficiency of 4.0% at a low power density of 60 W cm^−2^. It has been proven that the green UC emissions of the phosphor can be used as a ratio fluorescence thermometer, with a relative sensitivity of 1.3% K^−1^ at 293 K. However, due to poor chemical durability, especially in high-temperature or high-humidity environments, low laser-induced damage thresholds, and possible environmental pollution caused by fluorine sources, the application of fluoride is limited. Therefore, oxides with high physical, chemical, and thermal stability and low toxicity are still potential candidate materials despite relatively high phonon energy [16]. Wu et al. [17] synthesized a new oxide UC phosphor Ba_3_Y_4_O_9_:Er^3+^/Yb^3+^ for the first time through a high-temperature solid-state reaction. The optical temperature measurement performance of the phosphor was explored by characterizing the temperature-dependent FIR of its thermal coupling green and red UC emission bands. The results indicate that the green emissions and the red emissions have complementary temperature sensing ranges. The green emissions are suitable for temperatures above 350 K with a maximum sensitivity of 0.248% K^−1^ (563 K), while red emissions should be used for temperatures below 350 K with a maximum sensitivity of 0.371% K^−1^ (143 K). Zhang et al. [18] synthesized a series of K_3_Y(PO_4_)_2_ (KYP) phosphors doped with Yb^3+^-Er^3+^/Ho^3+^/Tm^3+^. The FIR of energy level transitions of Er^3+^(^2^H_11/2_,^4^S_3/2_→^4^I_15/2_ and ^4^F_9/2_→^4^I_15/2_), Ho^3+^(^5^F_4_,^5^S_2_→^5^I_8_ and ^5^I_5_→^5^I_8_), and Tm^3+^(^3^H_4_→^3^H_6_ and ^1^G_4_→^3^F_4_) were measured at different temperatures. The sensitivity of the materials was obtained through calculation, indicating that KYP:0.01Ho^3+^,0.2Yb^3+,^ and KYP:0.01Tm^3+^,0.2Yb^3+^ have the highest absolute sensitivity and relative sensitivity, respectively, while KYP:0.01Er^3+^,0.1Yb^3+^ shows the best comprehensive performance with absolute sensitivity of 0.304% K^−1^ (553 K) and relative sensitivity of 1.31% K^−1^ (239 K). Du et al. [19] prepared Yb^3+^/Er^3+^-doped Na_0.5_Bi_0.5_TiO_3_ (NBT) ceramics and enhanced the UC emission intensity through the addition of molybdenum. The temperature-dependent FIR of green UC emissions (525 nm and 550 nm) was studied in the range of 93–553 K. The maximum sensitivity of the material was found to be 0.35% K^−1^ at 493 K, indicating the potential application of Er/Yb/Mo triple-doped NBT ceramics in the field of optical temperature sensing. In addition to the above hosts, cubic silicate garnet Ca_3_Sc_2_Si_3_O_12_ (CSS) is widely used as a host for rare-earth-doped phosphors because of its excellent thermal and chemical stability and unique and multifunctional spectral properties. In previous research [20], we synthesized a CSS:Ce^3+^,Eu^2+^,Yb^3+^ phosphor with near-infrared (NIR) emission through the sol–gel combustion method. The energy transfer of Eu^2+^→Yb^3+^in CSS has been proven to be a unique resonant mechanism. The triply doped sample further enhanced the emission intensity of Yb^3+^ through Ce^3+^ and might be used to collect solar energy and convert it into NIR emissions of Yb^3+^. Sun et al. [21] reported Bi^3+^/Eu^3+^-doped CSS phosphors with blue/red emissions. Due to the high lattice energy of the CSS host, the CSS:Bi^3+^/Eu^3+^ phosphors showed excellent thermal stability. Therefore, the phosphors can be applied for anti-counterfeiting and fingerprint analysis at high temperatures. Wu et al. [22] developed a CSS:Ce^3+^,Na^+^ phosphor with a high internal quantum efficiency (85%) and co-fired it with glass powder to further prepare phosphor-in-glass (PiG) film with 508 nm cyan-green emissions and high thermal stability. Under the excitation of a blue laser, the complex of the CSS-PiG film and CaAlSiN_3_:Eu^2+^ obtained a color rendering index (Ra) up to 93, indicating that it was promising to be used in the field of laser illumination. Zhou et al. [23] reported an NIR CSS:Cr^3+^ phosphor, which exhibited an ultra-wide emission range of 650 to 900 nm under 460 nm excitation. The absorption of Cr^3+^ was improved by using Ce^3+^ as a sensitizer. Under 350 mA current driving, the NIR LED prepared with CSS:0.06Ce^3+^,0.03Cr^3+^ and a 450 nm blue chip achieved an output power of 21.65 mW and demonstrated excellent human tissue penetration ability. However, the temperature sensing performance of a Yb^3+^- and Er^3+^-doped Ca_3_Sc_2_Si_3_O_12_ phosphor has not been reported and brought to attention so far.

In this study, a CSS:Yb^3+^,Er^3+^ phosphor was prepared by the sol–gel combustion method. The lattice structure, UC emission characteristics, and temperature sensing performance under the 980 nm excitation of the samples were studied in detail. By monitoring the temperature-dependent luminescence behaviors of the ^2^H_11/2_ and ^4^S_3/2_ energy levels of Er^3+^, it was found that the maximum absolute sensitivity of the CSS:Yb^3+^,Er^3+^ phosphor is 0.67% K^−1^ at 500 K and the maximum relative sensitivity is 1.34% K^−1^ at 300 K based on FIR technology, which can be compared with the reported Yb^3+^- and Er^3+^-doped phosphors. The results show that the novel CSS:Yb^3+^,Er^3+^ phosphor has great potential for application in optical temperature sensors.

## 2. Materials and Methods

A series of Ca_2.96 − 2*x*_Yb*_x_*Er_0.02_Na_0.02 + *x*_Sc_2_Si_3_O_12_(*x* = 0, 0.01, 0.02, 0.05, 0.1, 0.2) phosphors were synthesized by sol–gel combustion method. Since the charge is unbalanced when Yb^3+^ and Er^3+^ ions replace Ca^2+^ ions, Na^+^ ions with the same concentration are used as charge compensation agents. First, the metal nitrate solutions were obtained by dissolving rare earth oxides (Yb_2_O_3_, 99.99%, Er_2_O_3_, 99.99%, and Sc_2_O_3_, 99.99%) with nitric acid. The stoichiometric reaction raw materials, including the metal nitrate solutions, Ca(NO_3_)_2_·4H_2_O(A.R.), and Na_2_CO_3_(A.R.), were added to an evaporating dish. Appropriate amounts of urea and ethanol were added to the reaction solution and heated to 65 °C on a heating agitator, evaporated overnight to remove excess water until a transparent sol was obtained. After that, the temperature was increased to 95 °C and maintained for several hours to obtain a dry gel. The obtained gel was heated from room temperature to 700 °C in the air in a muffle furnace at a heating rate of 5 °C/min and kept for 3 h at 700 °C and then naturally cooled to room temperature to obtain the precursor. After being ground, the precursor was transferred to a corundum crucible. It was heated from room temperature to 1400 °C in the air at a heating rate of 5 °C/min and kept for 6 h at 1400 °C. After naturally cooling to room temperature, the synthesized product was ground into powder and collected for further measurements. SYLGARD 184 (Dow Corning, including the base components and a curing agent with a weight ratio of 10:1) was stirred with the 5 wt% CSS:0.2Yb^3+^,0.02Er^3+^ phosphor at room temperature for 30 min and then heated in an oven at 80 °C for 1 h to encapsulate the CSS:0.2Yb^3+^,0.02Er^3+^ phosphor in solidified poly(dimethylsiloxane) (PDMS).

Powder X-ray diffraction (XRD) measurements were performed using a Rigaku D-max 2200 X-ray diffractometer (Rigaku Corporation, Tokyo, Japan) with Cu K*α* radiation at 40 kV and 26 mA. Diffuse reflectance spectroscopy (DRS) was measured by a Cary 5000 UV Vis NIR spectrophotometer (Agilent, Santa Clara, CA, USA) produced by the Varian company in the United States, and BaSO_4_ was used as the standard.

The morphology of the as-prepared sample was observed by scanning electron microscopy (SEM, Quanta 400 F, FEI Company, Hillsboro, OR, USA). Transmission electron microscopy (TEM) and element mapping analysis were carried out on a Tecnai G2 F30 instrument (FEI Company, Hillsboro, OR, USA). The photoluminescence emission (PL) spectrum was measured by a FLS 980 time-resolved and steady-state fluorescence spectrometer (Edinburgh Instruments, Livingston, UK). The light source was a 980 nm laser, and the temperature controller was an Oxford OptistatDN liquid nitrogen temperature control system (Oxford Instruments, Oxford, UK). The temperature measurement of the CSS:0.2Yb^3+^,0.02Er^3+^/PDMS composite was carried out by an infrared thermal imager (FLIR ONE Pro, Teledyne FLIR, Wilsonville, OR, USA) and fiber spectrophotometer (QE pro, Ocean Insight, Orlando, FL, USA).

## 3. Results and Discussion

As shown in Figure 1a, the crystal structure of Ca_3_Sc_2_Si_3_O_12_ belongs to a cubic crystal system with the space group *Ia*−3*d* (No. 230). The lattice parameters are *a* = 12.25 Å, *V* = 1838.3 Å^3^, and Z = 8. Each Ca^2+^ ion and the eight O^2−^ ions form a twisted dodecahedron. The bond lengths of four long and four short Ca−O bonds in the dodecahedron are 2.5660 (14) Å and 2.4324 (11) Å, respectively. Each Sc^3+^ ion and the six O^2−^ ions form an octahedron with the Sc−O bond length of 2.1062 (15) Å. The doped Er^3+^ and Yb^3+^ ions will replace the position of 8−fold Ca^2+^ ions because of the similar ionic radii (Ca^2+^: 1.12 Å, Er^3+^: 1.004 Å, Yb^3+^: 0.985 Å).

The powder XRD patterns of the host, Yb^3+,^ and Er^3+^ singly doped and co-doped CSS samples are shown in Figure 1b. The diffraction patterns of all samples are in good agreement with the standard reference data of Ca_3_Sc_2_Si_3_O_12_ (ICDD #72−1969), and no obvious impurity phase is observed, indicating that Yb^3+^ and Er^3+^ are successfully doped into the lattice of Ca_3_Sc_2_Si_3_O_12_ without obvious changes in the crystal structure.

According to the SEM image (Appendix A), the particle size of the material is approximately 20 μm. The high-resolution TEM image of the sample is shown in Figure 1c, and the clear lattice fringes indicate the high crystallinity of the sample. The measured d-spacing value of the (321) plane is 0.3202 nm, which is very close to the theoretical value of 0.3274 nm. The element mapping images of the CSS:0.2Yb^3+^,0.02Er^3+^ sample are shown in Figure 1d–j. It can be observed that all the elements are evenly distributed in the sample particle, which further confirms the successful doping of Yb^3+^ and Er^3+^ in the CSS host.

Figure 2a shows the UV-vis diffuse reflectance spectra of CSS:0.2Yb^3+^,0.02Er^3+^. There is a strong absorption band at about 980 nm, which is mainly attributed to the ^2^F_7/2_→^2^F_5/2_ transition of Yb^3+^. The absorption peaks at 380, 449, 486, 523, 647, and 1523 nm are caused by the electronic transitions from ^4^I_15/2_ to ^4^G_11/2_, ^4^F_3/2_, ^4^F_7/2_, ^2^H_11/2_, ^4^F_9/2_, and ^2^I_13/2_ of Er^3+^, respectively.

The UC emission spectra of the CSS:*x*Yb^3+^,0.02Er^3+^(*x* = 0, 0.01, 0.02, 0.05, 0.1, and 0.2) samples excited by the 980 nm laser are shown in Figure 2b. The emission peaks at 522, 555, and 658 nm are attributed to the characteristic transitions of ^2^H_11/2_, ^4^S_3/2,_ and ^4^F_9/2_→^4^I_15/2_ of Er^3+^, respectively. With the increase in the Yb^3+^ concentration, the UC emission of the sample is obviously enhanced. This is because the absorption cross-section of Yb^3+^ is larger than that of Er^3+^ in the near-infrared region around 980 nm, which can absorb more excitation energy and transfer it to Er^3+^. In addition, there is no concentration quenching in the concentration range of this study.

As shown in Figure 2c, with the increase in Yb^3+^ concentration, the green and red emissions of the CSS:*x*Yb^3+^,0.02Er^3+^ samples monotonically increase, while the red/green (R/G) ratio also shows an upward trend, which is similar to the reports of other Yb^3+^/Er^3+^ co-doped phosphors [24,25,26]. Due to the cross-relaxation (CR) of ^2^H_11/2_/^4^S_3/2_ (Er^3+^) + ^2^F_7/2_ (Yb^3+^)→^4^I_13/2_ (Er^3+^) + ^2^F_5/2_ (Yb^3+^) (Appendix A), the green UC emission is inhibited. Conversely, the population at the ^4^I_13/2_ state is increased by this CR process and the red emission is then enhanced through energy transfer, as discussed in detail in Figure 2f. Therefore, the R/G ratio rose with the increase in the Yb^3+^ concentration.

In order to determine the mechanism of UC luminescence, the UC emission spectra of the CSS:0.2Yb^3+^,0.02Er^3+^ sample at different pump power values were characterized, as shown in Figure 2d. Obviously, the UC emission intensity of the sample increases with the increase in the pump power. As a nonlinear process, the emission intensity (*I*) of the UC emission should increase in proportion to the pump power (*P*) of the excitation source, which can be expressed as [27,28,29]: (1)$$I\propto P^{n}$$
where *n* is the number of pump photons required to excite the luminescence center from the ground state to the excited state. Figure 2e shows the natural logarithm curves of the integrated emission intensity of the green emission (510–575 nm) and the red emission (630–695 nm) of the CSS:0.2Yb^3+^,0.02Er^3+^ sample at different pump power values. The curves were linearly fitted, and the *n* values were 1.68 and 1.42, respectively, indicating that the UC emissions of the phosphor are all derived from the two-photon process.

To describe the UC luminescence mechanism, the schematic diagram of the energy levels of Yb^3+^ and Er^3+^ and the possible UC two-photon process are shown in Figure 2f. The UC excitation process includes ground state absorption (GSA), excited state absorption (ESA), and energy transfer (ET). Because the absorption cross-section of Yb^3+^ in the near-infrared region is much larger than that of Er^3+^, both the green and red emissions are mainly realized by the effective ET process from Yb^3+^ to Er^3+^, and the contribution of GSA and ESA is small and can be ignored. Firstly, Yb^3+^ ions are excited from the ground state ^2^F_7/2_ to the excited state ^2^F_5/2_ by absorbing 980 nm photons. Yb^3+^ ions then relax to the ground state without radiation and transfer energy to nearby Er^3+^ ions. After obtaining energy, the Er^3+^ ions are excited from the ground state ^4^I_15/2_ to the excited state ^4^I_11/2_ [ET1 process: ^4^I_15/2_(Er^3+^)+^2^F_5/2_(Yb^3+^)→^4^I_11/2_(Er^3+^)+^2^F_7/2_(Yb^3+^)]. Subsequently, the Er^3+^ ions at the ^4^I_11/2_ excited state can relax to the ^4^I_13/2_ state without radiation or be further excited to the ^4^F_7/2_ state through energy transfer [ET2 process: ^4^I_11/2_(Er^3+^)+^2^F_5/2_(Yb^3+^)→^4^F_7/2_(Er^3+^)+^2^F_7/2_(Yb^3+^)]. After nonradiative relaxation of the Er^3+^ ions from the ^4^F_7/2_ excited state to the ^2^H_11/2_ and ^4^S_3/2_ states, 522 nm and 555 nm green emissions are generated through the radiative transitions of ^2^H_11/2_→^4^I_15/2_ and ^4^S_3/2_ →^4^I_15/2_, respectively. There are two possible processes for the red emission (658 nm) from the ^4^F_9/2_→^4^I_15/2_ transition: (1) the ^4^S_3/2_ excited state of Er^3+^ relaxes to the ^4^F_9/2_ state without radiation or (2) the ^4^I_13/2_ state of Er^3+^ is excited to the ^4^F_9/2_ state through energy transfer [ET3 process: ^4^I_13/2_(Er^3+^)+^2^F_5/2_(Yb^3+^)→^4^F_9/2_(Er^3+^)+^2^F_7/2_(Yb^3+^)] [30,31].

To explore the temperature sensing characteristics of the prepared sample, the temperature-dependent UC emission spectra normalized at 555 nm of CSS:0.2Yb^3+^,0.02Er^3+^ in the range of 300–500 K under 980 nm excitation was measured, as shown in Figure 3a. The corresponding contour map is illustrated in Figure 3b. Obviously, the position of the emission peaks of CSS:0.2Yb^3+^,0.02Er^3+^ does not change with the increase in temperature. However, the emission intensity at 522 nm increases significantly with increasing temperature.

Therefore, according to the Boltzmann distribution law, the FIR of the two thermally coupled levels ^2^H_11/2_ and ^4^S_3/2_ can be expressed as [32,33,34]: (2)$$FIR=\frac{I_{H}}{I_{S}}=\frac{N_{H}}{N_{S}}=\frac{g_{H}\omega_{H}A_{H}}{g_{S}\omega_{S}A_{S}}exp\left(-\frac{\Delta E}{kT}\right)=Bexp\left(-\frac{\Delta E}{kT}\right)$$
where *B* = *g_H_ω_H_A_H_*/*g_s_ω_s_A_s_*, the subscripts *H* and *S* represent the thermally coupled ^2^H_11/2_ and ^4^S_3/2_ energy levels, respectively, and *I*, *N*, *ω*, *g*, and *A* are defined as the UC emission intensity, population number, frequency, degeneracy, and spontaneous radiation transition rate, respectively. *k* is the Boltzmann constant. Δ*E* is the energy gap between the ^2^H_11/2_ and ^4^S_3/2_ states, and *T* is the absolute temperature. Equation (2) can also be simplified to the form of a linear equation, as shown below: (3)$$Ln\left(FIR\right)=LnB+\left(-\frac{\Delta E}{kT}\right)\;$$

Figure 3c shows the relationship between the Ln(FIR) and 1/*T* of the CSS:0.2Yb^3+^,0.02Er^3+^ in the temperature range of 300–500 K. The fitting degree of the experimental data is 0.9993, indicating a good linear fitting result. The slope of the fitted line is −Δ*E*/*k* = −1209.3. Therefore, the calculated Δ*E* = 840 cm^−1^, which is in line with the theoretical energy gap between the ^2^H_11/2_ and ^4^S_3/2_ energy levels of Er^3+^. Figure 3d shows the change in FIR with different temperatures. The coefficient *B* value obtained by fitting the experimental data is 15.63.

Absolute sensitivity (*S*_a_) and relative sensitivity (*S*_r_) are two important parameters that characterize the temperature measurement capability of a material, which are defined as follows [35,36]: (4)$$S_{a}=\left|\frac{d\left(FIR\right)}{dT}\right|=FIR\frac{\Delta E}{kT^{2}}$$
(5)$$S_{r}=\left|\frac{1}{FIR}\frac{d\left(FIR\right)}{dT}\right|=\frac{\Delta E}{kT^{2}}$$

As shown in Figure 3e, *S*_a_ gradually increases with the increasing temperature, and the maximum value is determined to be 0.67% K^−1^ at 500 K. However, $S_{r}$ shows a monotonous downward trend with the increasing temperature, and the maximum value is determined to be 1.34% K^−1^ at 300 K. By comparing *S*_a_ and *S*_r_ with other temperature sensing materials doped with Yb^3+^ and Er^3+^ (Table 1), we can find that CSS:0.2Yb^3+^,0.02Er^3+^ phosphor has superior temperature sensing performance and potential application prospects. The minimum temperature resolution obtained with this experimental setup is 1.03 K at 300 K, as shown in Appendix A, indicating that it still needs further optimization.

In addition, the temperature sensor must also have excellent repeatability, so that it can be used repeatedly without performance degradation. Figure 3f shows the temperature dependence of the FIR values of CSS:0.2Yb^3+^,0.02Er^3+^ for five heating–cooling cycles. It can be observed that the FIR values are almost completely reversible during repeated heating (300→500 K) and cooling (500→300 K) processes, indicating the excellent thermal repeatability of the CSS:Yb^3+^,Er^3+^ phosphor.

To verify the actual temperature measurement performance, the CSS:0.2Yb^3+^,0.02Er^3+^ phosphor was encapsulated in PDMS and heated and simultaneously measured by FIR technology and infrared thermal imaging (IRT) temperature measurement technology, as shown in Figure 4a. The results measured by IRT technology are shown in Figure 4b. The emission spectra of the phosphor under the corresponding temperature and 980 nm excitation are shown in Figure 4c. The FIR values were obtained from the spectra and the corresponding temperatures were calculated using Equation (3). The results measured by these two techniques at different temperature points were compared, as shown in Figure 4d. The temperature difference in the range of 300–350 K did not exceed 3.45 K, indicating that the CSS:0.2Yb^3+^,0.02Er^3+^ phosphor had an accurate temperature measurement capability. The temperature difference in the range of 350–425 K increased, probably because the heater heated up too quickly and the two techniques had different measurement speeds. In addition, the IRT technique might have a higher measurement error at high temperatures.

## 4. Conclusions

In summary, a series of CSS:*x*Yb^3+^,0.02Er^3+^ phosphors with excellent UC luminescence properties were synthesized by the sol–gel combustion method. Under 980 nm laser excitation, the green emission bands at 522 and 555 nm and the red emission band at 658 nm in the emission spectrum of the CSS:*x*Yb^3+^,0.02Er^3+^ phosphors can be observed. In addition, Yb^3+^ can effectively sensitize Er^3+^ to increase the emission intensity, and the energy transfer mechanism was revealed. The temperature sensing behavior of the CSS:0.2Yb^3+^,0.02Er^3+^ phosphor was studied by FIR technology. The results show that the maximum absolute sensitivity is 0.67% K^−1^ at 500 K and the relative sensitivity is 1.34% K^−1^ at 300 K. The temperature measurement performance of FIR technology using the CSS:Yb^3+^,Er^3+^phosphor was comparable to that of IRT technology, indicating that this material is expected to be used in optical thermometry.

## Figures and Tables

**Figure 1 nanomaterials-13-01910-f001:**
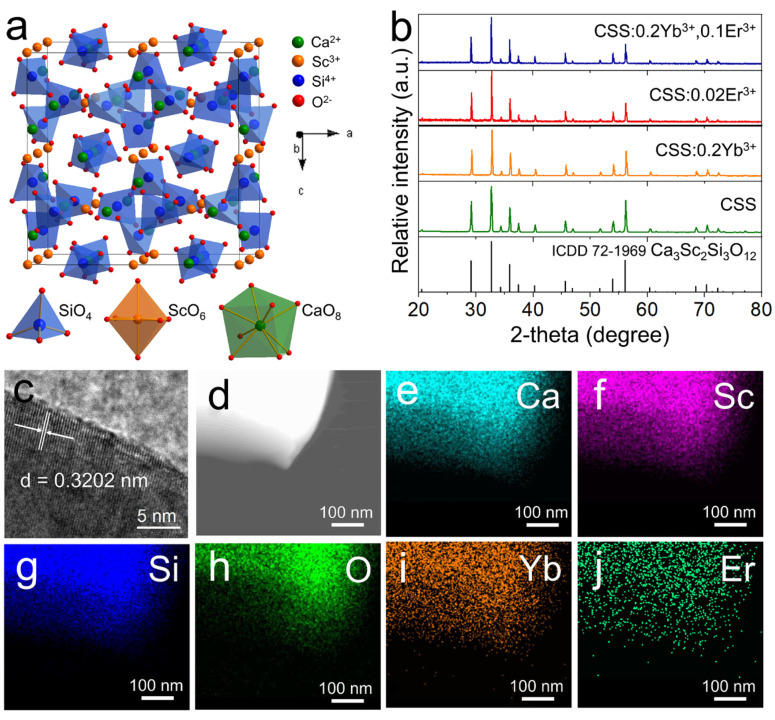
(**a**) Crystal structure of Ca_3_Sc_2_Si_3_O_12_. (**b**) The XRD patterns of the prepared CSS host, CSS:0.2Yb^3+^, CSS:0.02Er^3+,^ and CSS:0.2Yb^3+^,0.1Er^3+^. (**c**) High−resolution TEM images of CSS:0.2Yb^3+^,0.02Er^3+^. (**d**−**j**) TEM images of CSS:0.2Yb^3+^,0.02Er^3+^ and corresponding elemental mappings of Ca, Sc, Si, O, Yb, and Er.

**Figure 2 nanomaterials-13-01910-f002:**
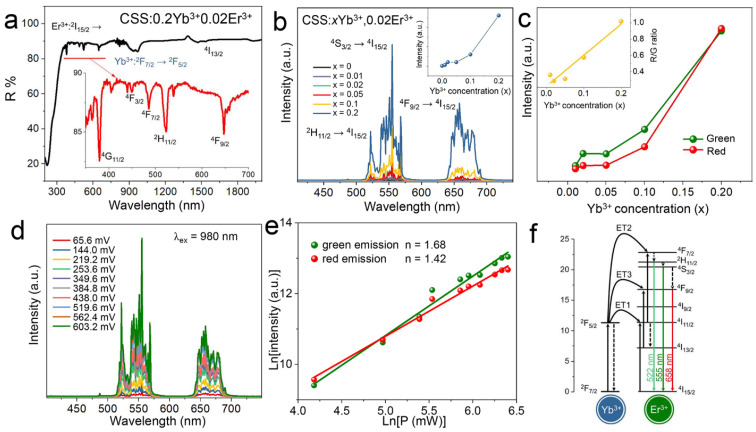
(**a**) Diffuse reflection spectra of CSS:0.2Yb^3+^,0.02Er^3+^. (**b**) UC emission spectra of CSS:*x*Yb^3+^,0.02Er^3+^ (*x* = 0–0.2). Inset: the corresponding concentration-dependent integrated luminescence intensity (510–695 nm). (**c**) The variation tendency of the integrated luminescence intensity of the green emission (510–575 nm) and the red emission (630–695 nm) of Er^3+^. Inset: the corresponding concentration-dependent R/G ratio. (**d**) UC emission spectra of CSS:0.2Yb^3+^,0.02Er^3+^ upon the excitation of 980 nm at different power values. (**e**) The Ln(*I*) versus Ln(*P*) for the integrated green emission (510–575 nm) and red emission (630–695 nm) of Er^3+^. (**f**) Energy level diagram of Yb^3+^ and Er^3+^ ions as well as the UC processes under 980 nm excitation.

**Figure 3 nanomaterials-13-01910-f003:**
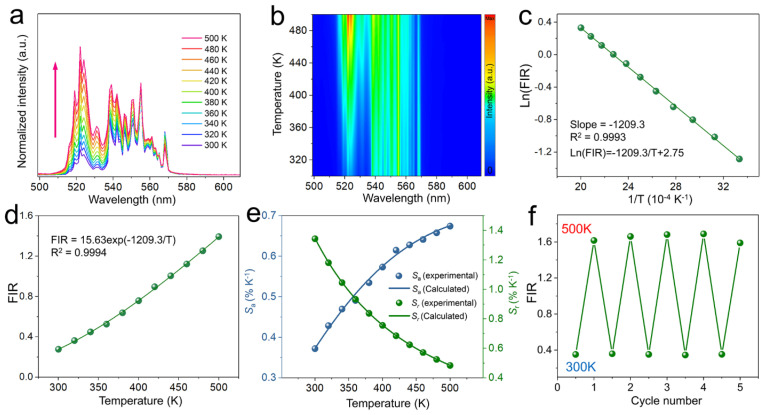
(**a**) Normalized temperature−dependent UC spectra and (**b**) contour plots of CSS:0.2Yb^3+^,0.02Er^3+^ under 980 nm excitation. Temperature dependence of (**c**) Ln(FIR), (**d**) FIR, and (**e**) sensitivity of CSS:0.2Yb^3+^,0.02Er^3+^ under 980 nm excitation. (**f**) Temperature cycling measurement of CSS:0.2Yb^3+^,0.02Er^3+^ between 300 and 500 K.

**Figure 4 nanomaterials-13-01910-f004:**
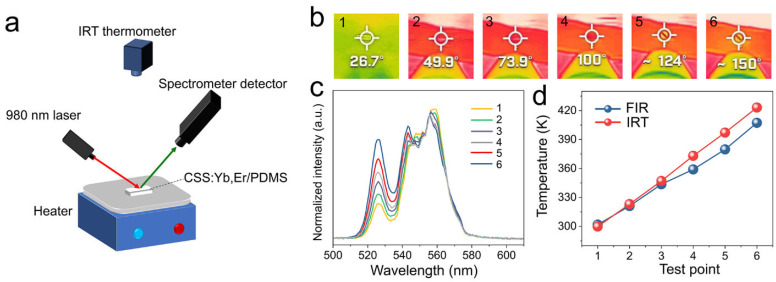
(**a**) Schematic diagram of IRT and FIR technology. (**b**) Measurement results of IRT technology at different temperature points (1−6). (**c**) Emission spectra of the CSS:0.2Yb^3+^,0.02Er^3+^/PDMS complex at different temperature points under 980 nm laser excitation. (**d**) Comparison of IRT and FIR technology measurement results.

**Table 1 nanomaterials-13-01910-t001:** The maximum *S_a_* and *S_r_* of various Yb^3+^, Er^3+^ co-doped phosphors using the FIR technique.

Phosphor	Temperature Range (K)	*S_a-_*_Max_(% K^−1^)	*S_r-_*_Max_ (% K^−1^)	Ref.
Ba_3_Y_4_O_9_:Yb^3+^,Er^3+^	83–563	0.248 (563 K)	/	[17]
K_3_Y(PO_4_)_2_:Yb^3+^,Er^3+^	293–553	0.304 (553 K)	1.31 (239 K)	[18]
Na_0.5_Bi_0.5_TiO_3_:Yb^3+^,Er^3+^	93–553	0.35 (493 K)	/	[19]
Gd_2_O_3_:Yb^3+^,Er^3+^	300–900	0.39 (300 K)	/	[37]
NaYF_4_:Yb^3+^,Er^3+^	293–753	0.39 (500 K)	1.3 (293 K)	[15]
Y_2_O_3_:Yb^3+^,Er^3+^	93–613	0.44 (427 K)	/	[38]
Al_2_O_3_:Yb^3+^,Er^3+^	295–973	0.51 (495 K)	/	[39]
LuVO_4_:Yb^3+^,Er^3+^@SiO_2_	303–353	0.572 (353 K)	1.173 (303 K)	[40]
Ba_2_In_2_O_5_:Yb^3+^,Er^3+^	303–573	0.65 (498 K)	/	[41]
Ca_2_MgWO_6_:Yb^3+^,Er^3+^	303–573	0.82 (453 K)	0.92 (303 K)	[42]
LuVO_4_:Yb^3+^,Er^3+^	303–423	0.82 (423 K)	1.12 (303 K)	[43]
CSS:Yb^3+^,Er^3+^	300–500	0.67 (500 K)	1.34 (300 K)	This work

## Data Availability

The data in this study are available from the corresponding author upon request.

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
