# Peer review of "The Upconversion Luminescence of Ca3Sc2Si3O12:Yb3+,Er3+ and Its Application in Thermometry"

_nanomaterials, 2023, doi:10.3390/nano13131910_

Round 1

Reviewer 1 Report

This is a fine paper that can be published after minor revision.

I have only some questions to be answered.

Figure 2b, inset. “The corresponding concentration-dependent luminescence intensity.” Please indicate the wavelength for which the dependence is obtained.”

Fig. 2c. Please indicate wavelengths of red and green emission for which the dependence is obtained.

Fig. 2e. Please explain how you estimated intensity of green emission (510-575 nm) and red emission (630-695 nm).

Line 86. “Diffuse reflectance spectroscopy (DRS) was measured by Cary 5000 UV Vis NIR spectrophotometer…” Please make correction as follows: “Diffuse reflectance spectra (DRS) were measured by Cary 5000 UV Vis NIR spectrophotometer…”

Line 187. Please correct the sentence: “The corresponding contour map are illustrated in Figure 3b.”

The quality of English Language is fine. I have only two comments:

Line 86. “Diffuse reflectance spectroscopy (DRS) was measured by Cary 5000 UV Vis NIR spectrophotometer…” Please make correction as follows: “Diffuse reflectance spectra (DRS) were measured by Cary 5000 UV Vis NIR spectrophotometer…”

Line 187. Please correct the sentence: “The corresponding contour map are illustrated in Figure 3b.”

Author Response

Authors general response: We greatly appreciate the careful reading of the manuscript by the Reviewer and the constructive and useful comments. These comments have been fully answered point by point.

Figure 2b, inset. “The corresponding concentration-dependent luminescence intensity.” Please indicate the wavelength for which the dependence is obtained.”

Authors response: Thanks for the good comment. The luminescence intensity is obtained by calculating the spectral integral area between 510 and 695 nm.

We have revised the figure annotation as follow:

Inset: the corresponding concentration-dependent integrated luminescence intensity (510-695 nm).

Fig. 2c. Please indicate wavelengths of red and green emission for which the dependence is obtained.

Authors response: Thanks for the good comment. We use spectral integral areas of 510-575 nm and 630-695 nm as the intensities of green and red emissions, respectively.

We have revised the figure annotation as follow:

(c) The variation tendency of the integrated luminescence intensity of green emission (510-575 nm) and red emission (630-695 nm) of Er3+.

Fig. 2e. Please explain how you estimated intensity of green emission (510-575 nm) and red emission (630-695 nm).

Authors response: Thanks for your comment. We calculated the spectral integral areas of 510-575 nm and 630-695 nm as the intensities of green and red emissions, respectively.

We have revised the figure annotation as follow:

(e) The Ln(I) versus Ln(P) for integrated green emission (510-575 nm) and red emission (630-695 nm) of Er3+.

Line 86. “Diffuse reflectance spectroscopy (DRS) was measured by Cary 5000 UV Vis NIR spectrophotometer…” Please make correction as follows: “Diffuse reflectance spectra (DRS) were measured by Cary 5000 UV Vis NIR spectrophotometer…”

Authors response: Thanks for your good suggestion. We have revised it.

Line 187. Please correct the sentence: “The corresponding contour map are illustrated in Figure 3b.”

Authors response: Thanks for your good suggestion. We have revised it as follow:"“The corresponding contour map is illustrated in Figure 3b.”"

Comments on the Quality of English Language

The quality of English Language is fine. I have only two comments:

Line 86. “Diffuse reflectance spectroscopy (DRS) was measured by Cary 5000 UV Vis NIR spectrophotometer…” Please make correction as follows: “Diffuse reflectance spectra (DRS) were measured by Cary 5000 UV Vis NIR spectrophotometer…”

Authors response: Thanks for your good suggestion. We have revised it.

Line 187. Please correct the sentence: “The corresponding contour map are illustrated in Figure 3b.”

Authors response: Thanks for your good suggestion. We have revised it as follow:"“The corresponding contour map is illustrated in Figure 3b.”"

Reviewer 2 Report

The article entitled "The up-conversion luminescence of Ca3Sc2Si3O12 :Yb3+,Er3+ and its application in thermometry" presents a study on the synthesis of luminescent scandium and calcium silicate powders used for thermometry. The article is well written in understandable English.

The objectives of the study are briefly addressed in the introduction, but there is no mention at all of other possible host matrices. A more complete bibliography, including the data in the table on page 7, would be helpful in understanding the place of the chosen matrix in the landscape. In addition, the reasons for the choice of this matrix are discussed very briefly. It would be desirable for them to be developed further. 

It should be made clear that these powders are not likely to be used in in vivo biology. If this were the case, it would be important to specify the possibilities for functionalizing the particle surfaces. Furthermore, the size of the particles is never mentioned. It would be interesting to provide this information.

The authors indicate that the powders are sintered at 1400°C in a corundum crucible. However, how did the authors ensure that no reactions occurred between the phase and the crucible? Corundum is not very chemically stable with respect to sodium and silicates.

The authors invoke their X-ray diffraction to ensure that the compounds are pure. However, these diagrams are far too small to judge. I am also surprised that there is no distortion of the lattice with doping, although this would be expected. Similarly, the inter-reticular distance measurement given on line 127 seems to indicate very little variation. Could this variation be linked to doping?

The authors indicate that the temperature difference in the 300-350K range does not exceed 3.45K. This seems to me to be quite high compared with other systems. Can the authors make a comparison on this point?

Reviewer 3 Report

The authors presented an interesting article on the upconversion and thermometric properties of calcium scandium silicate doped with Yb and Er. The presented results are new, the research design is correct and the article quality is very high. I have only minor issues regarding this manuscript. I recommend it is published after minor issues are addressed.

1.       The synthesis is described as sol-gel combustion, but the annealing consists of two stages: 700 C/ 3h and 1400 C/ 6h. I’m guessing the combustion occurs at the first stage – does the time of 3h include heating/cooling or does the combustion require such long time to occur? Some comment for clarity in the experimental section could be useful.

2.       What is the grain size of obtained material? High temperature annealing would typically grow the grains to micrometric size.

3.       How was the 2% Er concentration selected as optimal? Was there a preliminary study to be cited?

4.       Line 149 – “the energy of 4I13/2 state is saturated”. I believe more precise expression could be used in this sentence and the mentioned cross-relaxation would be more clear when presented on the figure.

5.       The thermal resolution above 1 K is quite poor. There are many 980 nm – pumped optical thermometers with better resolution according to DOI: 10.1016/J.JALLCOM.2021.162794. I think the authors should acknowledge that in the manuscript.

Author Response

Authors general response: We greatly appreciate the careful reading of the manuscript by the Reviewer and the constructive and useful comments. These comments have been fully answered point by point.

1.The synthesis is described as sol-gel combustion, but the annealing consists of two stages: 700 C/ 3h and 1400 C/ 6h. I’m guessing the combustion occurs at the first stage – does the time of 3h include heating/cooling or does the combustion require such long time to occur? Some comment for clarity in the experimental section could be useful.

Authors response: Thanks for the good comment. The time of 3h does not include the heating and cooling process. We have added the following details in the experimental section:

The obtained gel was heated from room temperature to 700℃ in the air in a muffle furnace at a heating rate of 5℃/min and kept for 3 hours at 700℃, then naturally cooled to room temperature to obtain the precursor. After being ground, the precursor was transferred to a corundum crucible. It was heated from room temperature to 1400℃ in air at a heating rate of 5℃/min and kept for 6 hours at 1400℃. After naturally cooling to room temperature, the synthesized product was ground into powder and collected for further measurements.

2.What is the grain size of obtained material? High temperature annealing would typically grow the grains to micrometric size.

Authors response: Thanks for the good suggestion. Indeed, the material have grown to micrometric size. According to the SEM image, the particle size of the material is approximately 20 μm. We have supplemented a description of the particle size of the material in the manuscript and added a SEM image to the Supporting Information:

According to the SEM image (Figure. S1), the particle size of the material is approximately 20 μm.

Figure S1. SEM image of CSS:0.2Yb3+,0.02Er3+.

3.How was the 2% Er concentration selected as optimal? Was there a preliminary study to be cited?

Authors response: Thanks for the good comment. According to existing research, a concentration of 1%-5% Er is more suitable, and many studies have chosen 2% as the doping concentration for Er. Such as the following literatures which have been cited in this article:

Chem. Eng. J. 2016, 297, 26-34. (doi.org/10.1016/j.cej.2016.03.149) YNbO4:x%Yb3+,2%Er3+

 J.Mater. Chem. C 2018, 6, 5453-5461.(doi.org/10.1039/C8TC01806E) NaYF4:20%Yb3+,2%Er3+

4.Line 149 – “the energy of 4I13/2 state is saturated”. I believe more precise expression could be used in this sentence and the mentioned cross-relaxation would be more clear when presented on the figure.

Authors response: Thanks for the good suggestion. We have revised the expression and added a schematic diagram of the cross-relaxation in Supporting Information:

Due to the cross-relaxation (CR) of 2H11/2/4S3/2 (Er3+)+ 2F7/2 (Yb3+) → 4I13/2 (Er3+)+ 2F5/2 (Yb3+) (Figure S2),the green UC emission is inhibited. Conversely, the population at 4I13/2 state is increased by this CR process and the red emission is then enhanced through energy transfer, as discussed in detail in Figure 2f. Therefore, R/G ratio is rose with the increase of Yb3+ concentration.

Figure S2. the cross-relaxation of 2H11/2/4S3/2 (Er3+)+ 2F7/2 (Yb3+) → 4I13/2 (Er3+)+ 2F5/2 (Yb3+)

5.The thermal resolution above 1 K is quite poor. There are many 980 nm – pumped optical thermometers with better resolution according to DOI: 10.1016/J.JALLCOM.2021.162794. I think the authors should acknowledge that in the manuscript.

Authors response: Thanks for the good comment. We have made revisions in the manuscript and Supporting Information.

Manuscript: The minimum temperature resolution obtained with this experimental setup is 1.03 K at 300 K, as shown in Figure S3, indicating that it is still needs further optimization.

Supporting Information: the calculated temperature resolution δT is shown in Figure. S3 and the minimum value of δT is 1.03 K at 300 K.

Round 2

Reviewer 2 Report

the authors answered the questions well.

I still think that the subject is not very original, but the work done deserves to be published.